# Expression Kinetics of Regulatory Genes Involved in the Vesicle Trafficking Processes Operating in Tomato Flower Abscission Zone Cells during Pedicel Abscission

**DOI:** 10.3390/life10110273

**Published:** 2020-11-06

**Authors:** Srivignesh Sundaresan, Sonia Philosoph-Hadas, Joseph Riov, Shoshana Salim, Shimon Meir

**Affiliations:** 1Department of Postharvest Science, Agricultural Research Organization (ARO), The Volcani Center, Rishon LeZion 7528809, Israel or srivignesh.horti@gmail.com (S.S.); vtsoniap@volcani.agri.gov.il (S.P.-H.); shoshi@volcani.agri.gov.il (S.S.); 2The Robert H. Smith Institute of Plant Sciences and Genetics in Agriculture, The Robert H. Smith Faculty of Agriculture, Food and Environment, The Hebrew University of Jerusalem, Rehovot 7610001, Israel; joseph.riov@mail.huji.ac.il

**Keywords:** abscission, flower abscission zone, GTPases, SNAREs, tomato, transcriptome, vesicle trafficking

## Abstract

The abscission process occurs in a specific abscission zone (AZ) as a consequence of the middle lamella dissolution, cell wall degradation, and formation of a defense layer. The proteins and metabolites related to these processes are secreted by vesicle trafficking through the plasma membrane to the cell wall and middle lamella of the separating cells in the AZ. We investigated this process, since the regulation of vesicle trafficking in abscission systems is poorly understood. The data obtained describe, for the first time, the kinetics of the upregulated expression of genes encoding the components involved in vesicle trafficking, occurring specifically in the tomato (*Solanum lycopersicum*) flower AZ (FAZ) during pedicel abscission induced by flower removal. The genes encoding vesicle trafficking components included soluble N-ethylmaleimide-sensitive factor attachment protein receptors (SNAREs), SNARE regulators, and small GTPases. Our results clearly show how the processes of protein secretion by vesicle trafficking are regulated, programmed, and orchestrated at the level of gene expression in the FAZ. The data provide evidence for target proteins, which can be further used for affinity purification of plant vesicles in their natural state. Such analyses and dissection of the complex vesicle trafficking networks are essential for further elucidating the mechanism of organ abscission.

## 1. Introduction

Organ abscission is a complex developmental process, which involves anatomical, physiological, biochemical, and molecular aspects [1,2]. Intensive research during the years has demonstrated that the abscission process proceeds through the four following stages, which finally lead to organ separation from the mother plant: (**A**) anatomical differentiation of cells to form the specific abscission zone (AZ) layer. (**B**) acquisition stage, during which the AZ cells become competent to respond to abscission signaling, thereby rendering the AZ ready for abscission execution. (**C**) execution stage, which leads to cell separation after the AZ cells were activated by ethylene, by increased synthesis and activation of cell wall hydrolyzing enzymes. (**D**) protection stage, which includes synthesis of extensible boundary and defense layers onto the surface of the retained portion of the AZ layer, and secretion of pathogenesis-related (PR) proteins [3,4,5,6].

Various proteins, which are involved in the degradation of the middle lamella and loosening of the cell wall, were identified and characterized in the AZs of several plants at stages C and D of the abscission process. The related proteins include cellulases (beta-1,4-endoglucanases, CELs), polygalacturonases (PGs), xyloglucan endotransglucosylases/hydrolases (XTHs), and expansins (EXPs) [5,6,7]. Additionally, the upregulation of defense genes, which include *PR* genes, is commonly observed in these stages of abscission [8,9,10,11]. The upregulation of *PR* gene expression during abscission has a role in protecting the abscising cells from pathogen invasion [8]. Transcriptome analyses of tomato flower and soybean leaf AZs, focusing on genes associated with disassembly of the cell wall, and genes linked to the biosynthesis of a new extracellular matrix, were recently performed [12]. These studies demonstrated the upregulation of AZ-specific genes associated with cell wall disassembly, including *CEL*s, *PG*s, and *EXP*s, as well as an early upregulation of genes related to the synthesis of a waxy-like cuticle involved in the formation of an extensible boundary layer on the surface of separating cells.

The proteins involved in cell wall disassembly, boundary layer formation, and protection layer assembly, as well as the metabolites required for carrying out these processes, have to be secreted to the separating cells in the AZ, as described in the models presented in several reports [7,13,14]. Scientists have been aware for a long time that there is a need for increased activity of the secretory machinery of the endomembrane system to enable the release of cell wall modifying enzymes to implement the abscission process [15,16].

The mechanisms of the essential process of protein secretion have been extensively studied, as recently reviewed by Wang et al. (2018) [17] and Cui et al. (2020) [18]. The conventional protein secretion (CPS) pathway from the endoplasmic reticulum (ER) to the Golgi apparatus is responsible for the delivery of proteins with an N-terminal leading sequence or a transmembrane domain, while the unconventional protein secretion (UPS) pathways operate from the cytosol to the cell exterior for the transport of leaderless secretory proteins. These UPS pathways are carried out by the fusion of membrane-bound vesicles or organelles with the plasma membrane (PM), including secretory multi-vesicular bodies (MVBs) and exocyst-positive organelles (EXPOs). Most of these pathways usually operate in response to pathogen attack and stress conditions [19,20,21,22]. Accumulation of vesicular structures, such as paramural bodies (PBs) and MVBs, was observed at the proximal side of the tomato leaf AZ and in the tomato flower AZ (FAZ) [23,24,25]. 

The transfer of materials between organelles is mediated by carrier vesicles that continually bud from the membrane and fuse with another one. It is generally believed that most (about 85%) of the protein secretion or exocytosis is achieved via the CPS pathway, operating from the ER to the trans-Golgi network (TGN), and to the PM in the plant endomembrane system [17,26,27,28]. This pathway involved in plant cell cytokinesis, cell wall synthesis, and metabolic systems has been well characterized and reviewed [29,30,31,32,33,34], but the trafficking mechanism in the abscission systems is poorly understood.

The processes of cell wall biosynthesis, as well as the delivery of transmembrane receptors, cell wall enzymes, soluble N-ethylmaleimide-sensitive factor attachment protein receptors (SNAREs), SNARE regulators, and other membrane proteins to the PM, operate through the Golgi apparatus and the post-Golgi trafficking pathways [24,33,35]. Basically, each vesicle transport reaction proceeds through four essential steps, including vesicle budding, transport, tethering, and fusion [35,36,37]. To ensure vesicles delivery between specific donor and acceptor compartments, these four steps are tightly regulated [37,38,39,40]. 

SNARE proteins comprise of three conserved families of membrane-bound proteins, synaptobrevin/VAMP, syntaxin, and SNAP-25/light chain families, which act in the late stages of the events leading to bilayer fusion. The SNAREs confer the tight docking, and probably also the subsequent fusion of membrane bilayers, since they act downstream of both tethering factors and Rab GTPases, which lead to loose membrane attachment. Membrane fusion and the delivery of cargo is driven by the four-helix bundle (trans-SNARE complex), assembled from the v- (vesicle) and t- (target) SNARE members of the above mentioned three families of membrane-bound proteins [27,33,35,36,37]. Rab GTPases, tethering complexes, and AAA-type ATPases are the proteins necessary to confer fusion between the donor and the acceptor compartments [40], as well as SNARE regulators. We adopted the classification of the two groups of SNARE regulators according to the review of Gerst (2003) [38], which include matchmakers that facilitate the assembly of cognate SNAREs, and match-breakers that keep SNAREs in a conformation-inactive state. 

The reports that relate vesicle trafficking to abscission systems are limited, although this process is very important for the execution of organ abscission. Few studies showed changes in the expression of genes that regulate the process of vesicle trafficking during abscission. Agusti et al. (2012) [13] observed the upregulation of several genes involved in vesicle trafficking, such as *SNARE-like protein* and *Syntaxin*, in the laminar AZ of abscising citrus leaves following a cycle of water stress/rehydration. These authors also identified the transportation of vesicles containing cell wall modifying enzymes to the middle lamella as the main biological process activated in the laminar AZ during the early steps of rehydration-promoted leaf abscission following water stress.

Analysis of gene expression profiles in the AZ of melon [14] and mature olive [41,42] fruits revealed that the sequential induction of genes encoding cell wall modifying enzymes was associated with the upregulation of genes encoding Rab-GTPases, small GTPases, and SNAREs involved in endo- and exo-cytosis during natural abscission of mature fruits. Activation of vesicle trafficking involving small GTPases was also found to be required for cell wall modification during abscission of Arabidopsis floral organs [43,44,45]. The differentially expressed miRNAs involved in the formation of AZ cells was investigated, and two sRNA libraries were constructed using AZ tissues of cotton pedicels treated with ethephon or water [46]. This research revealed seven novel miRNAs, which were differentially expressed in the AZ tissues, and some of their target genes were found to operate as GTPase activators. A recent report of Merelo et al. (2019) [47] demonstrated that 25 genes, associated with vesicle trafficking, were specifically regulated in the calyx AZ (AZ-C) of detached mature orange fruits following a 24-h exposure to ethylene. These genes included two members of the RabD-GTPases involved in the ER to Golgi transport, two RabA-GTPases involved in exocytosis, a small GTPase of the ARF family, and a *CitSYP131* SNARE localized on the PM. In contrast, some genes involved in endocytosis were specifically downregulated by ethylene in the AZ-C cells. The authors suggested that while the exocytosis secretory pathway became operative in the AZ-C cells during fruit abscission, the secretory pathway to the vacuole might be impaired during this process. 

Recently, our group reported that the Tomato Hybrid Proline-Rich Protein (THyPRP) plays a major role in tomato flower pedicel abscission by regulating the competence of the FAZ cells to respond to ethylene signaling [48,49]. Among the various genes whose expression was changed following abscission induction in this system, THyPRP was also found to regulate the induced expression of several exocytosis-related genes, such as *Serpins*, *Syntaxins*, and *SNARE*-*like* genes that are necessary for cell separation. This study further demonstrates the involvement of vesicle trafficking components in a flower abscission system.

Although, as described above, there are few studies which investigated the genes that regulate the process of vesicle trafficking in various abscission systems, our knowledge of this topic is still limited, mainly because the reported data are related only to one or two time points during the abscission process. The aim of the present study was to further investigate in details the kinetics of the expression changes of genes encoding the three components involved in mediating vesicle trafficking between organelles and the PM in the tomato FAZ during the abscission process. For this purpose, we used a tomato AZ-specific microarray chip [50] to examine the gene expression changes at the early stages of pedicel abscission execution (preparation for abscission execution—late stage B), and at stages C and D. This is the first report showing a detailed kinetics of the expression of vesicle trafficking genes in the FAZ during all the stages of organ abscission, using the model system of tomato pedicel abscission. These results provide a clear picture of how the processes of protein secretion by vesicle trafficking are regulated, programmed, and orchestrated in the FAZ.

## 2. Results and Discussion

The experimental design (i.e., the sampling timing of the tissues for the analysis of gene expression) of the present study was based on our previous article [12], which described the kinetics of the tomato flower pedicel abscission following flower removal. No pedicel abscission could be observed at the first 8 h after flower removal in the tomato cv. “New Yorker”. However, 12 and 16 h after flower removal, pedicel abscission rates were 20% and 80%, respectively, and abscission was completed after 20 h. This study demonstrated the expression of genes encoding for the synthesis of enzymes and metabolites that constitute the cargo required to reach the cell wall and the inter-cellular space. Based on the above observations, we defined the period of 0–8 h after flower removal as the abscission stage B, the period of 8–16 h as the abscission stage C, and the period of 16–20 h as the abscission stage D for this tomato cultivar. However, it should be noted that according to the expression profiles of the genes related to cell wall disassembly, PR, proteinaceous extracellular matrix, and synthesis of a waxy cuticle, there is some overlapping between these abscission stages in the tomato FAZ [12]. 

The secretion of cargo molecules, gene products, and metabolites to the cell wall of the separation layer in the FAZ should be synchronized with gene expression and translation both in timing and location, for ensuring that the cargo molecules will be transported only to the correct target compartment. In the present study, our microarray results generated 110 *GTPase-related* and 82 *SNARE* and *SNARE regulators* genes, expressed in the FAZ and/or in the non-AZ (NAZ). The expression pattern of these genes in the tomato FAZ and NAZ at various time points after flower removal are presented in Figure 1 and Figure 2, and Table 1, Table 2, Table 3 and Table 4, and their heat maps are presented in Appendix A. The complete details of the gene nomenclature, expression timing, and annotation are presented in the Appendix A. To validate the microarray assay results, we selected five genes, and analyzed their expression by qRT-PCR. The qRT-PCR results were similar to the microarray results (Appendix A). In order to examine the regulation of vesicle trafficking involved in tomato pedicel abscission, we selected genes that were specifically upregulated in the FAZ starting at 4 h (Figure 1) or at 8, 12, and 16 h (Figure 2) after flower removal, and presented their expression levels according to their kinetics and functional groups.

The protein functions and their specific intracellular compartments presented in Table 1, Table 2, Table 3 and Table 4 are based on transient expression assays, using green florescent protein (GFP) fused proteins according to published procedures [28,29,38,51,52]. The t- or v-SNARE definition, and the Qa-Qb-R- and Qc-SNARE motifs are based on published data [29,34,51]. 

The results presented in Figure 1 and Table 1 demonstrate that 4 h after flower removal, 27 vesicle trafficking-related genes were significantly upregulated in the FAZ, including 12 *GTPase-related*, 8 *SNAREs*, and 7 *SNARE regulators* genes. The expression of most of these genes further increased during the following sampling time points, or remained high up to 16 h after flower removal, when 80% of the pedicels had already abscised. At 20 h after flower removal, when the pedicel abscission process was completed, the expression of some of these genes decreased to the expression level observed at zero time.

The following genes were upregulated at 4 h after flower removal: (**a**) genes that regulate vesicle budding-two *Arf GTPase* (Figure 1I-C) and four *Dynamin GTPase* (Figure 1I-B) genes (Figure 1, and Table 1). (**b**) genes that regulate protein trafficking from the ER to the *cis*-Golgi-four *Rab GTPase* (Figure 1I-A), *Sar GTPase* (Figure 1I-D), and *Obg GTPase* (Figure 1I-E) genes. (**c**) genes that regulate protein trafficking from the ER through the *cis*-Golgi and the TGN to the PM—eight *SNARE* genes (seven *t-SNAREs* and one *v-SNARE*) (Figure 1II), as well as six matchmaker SNARE regulators (Figure 1III), and one match-breaker SNARE regulator (Figure 1IV). The expression levels of most of the genes that were upregulated at 4 h after flower removal were similar to their expression levels at 8 h, and the expression levels of some genes decreased from 8 h to 12 h. All the genes listed in Table 3 were still highly expressed in the FAZ compared to their expression in the NAZ at these time points. These results demonstrate that all the network of vesicle trafficking through the conserved pathway, from the ER to the PM, was specifically activated in the FAZ already at the early stage of the pedicel abscission process. 

In addition to the 27 vesicle trafficking-related genes that were upregulated at 4 h after flower removal and remained highly expressed in the FAZ, 15 more vesicle trafficking-related genes were upregulated at 8 h after flower removal, when cell separation started. The upregulated genes included genes encoding six Rab-, Ras- and Ran-GTPases, four t-SNAREs, three v-SNAREs, and two matchmaker SNARE regulators (Figure 2I–III, Table 2). These genes probably reinforce the transportation of new cargo molecules into the exocytosis pathway. At 12 h after flower removal, five additional vesicle trafficking-related genes exhibited increased expression (Figure 2IV–VI, Table 3). At 16 h after flower removal, four more vesicle trafficking-related genes were upregulated (Figure 2VII,VIII). Taken together, the above results demonstrate the changes in the expression kinetics of vesicle trafficking-related genes, which occurred during the 20-h pedicel abscission process following flower removal. These results suggest that genes that were upregulated at different time points during the abscission process are responsible for trafficking of various cargo molecules to different locations of the exocytosis pathway from the ER to the PM. 

The *SNARE SYP121* (*SYR1/PEN1*) and *SNAP33* genes were concomitantly upregulated in the tomato pedicel FAZ during pedicel abscission (Figure 1II-3,II-5). In Arabidopsis, *SYP121* and its close homologue, *SYP122*, were ubiquitously expressed in the PM throughout plant development [52,53]. SNARE SYP121, as a member of the SNARE complex, drives vesicle fusion to the PM late in the secretory pathway, together with SNAP33 and the functionally redundant v-SNAREs [54]. Additionally, SYP121 promoted the gating of the inward-rectifying K^+^ channel, thus coordinating K^+^ uptake, which is a vital step for increased cell volume and turgor control [55]. In this regard, it should be noted that the elongation and increased volume of the AZ cells at the fracture plane, which are driven by the turgor pressure, are essential for the cell separation process, as recently reviewed [56]. Accordingly, we suggest a possible additional role for SYP121 in enabling increased elongation and volume of the FAZ cells during the abscission cell separation process, in addition of being a part of the SNARE complex which operates in the vesicle trafficking network.

The main novelty of this report is expressed by presenting, for the first time, detailed kinetics of the expression of vesicle trafficking genes during all stages of the abscission process, which can be correlated with the expression of different components of cargo-related genes presented in a previous article [12]. Up to now, some reports presented a description of the expression of genes related to vesicle trafficking in the AZ, but their analysis was performed only during one or two stages of organ abscission. For example, analysis of gene-expression in the AZs of mature olive fruits revealed, that genes involved in vesicle trafficking were upregulated at 217 days post anthesis (DPA), when natural fruit abscission occurred. These genes encoded small GTPases, seven Rab-GTPases, five Arf-like GTPases, two Ran GTPases, and one Rho GTPase, three Syntaxin/SNAREs, five dynamins, and eight V-type ATPases [41]. Recently, the above group analyzed the expression of genes related to olive fruit ripening and abscission, demonstrating that different and specific cell wall genes were upregulated in the fruit pericarp and in the AZ tissues, and that the secretion regulation of the related proteins involved different Rab-GTPases and syntaxin encoding genes [42]. 

Another report dealing with vesicle trafficking during organ abscission described a sharp increase in the ethylene production rate in melon fruits, which occurred during 36 to 38 DPA, thereby inducing fruit abscission that occurred at 42 DPA [14]. Analysis of gene expression in the AZ of mature melon fruits demonstrated a sequential induction of genes encoding cell wall degrading enzymes in parallel with genes involved in endo- and exocytosis (*small GTPase* and *SNAREs*), most of which were upregulated between 36 to 38 DPA, when cell separation between the pedicel and the fruit was visible [14]. The expression of these genes decreased between 38 to 40 DPA, when abscission was almost completed, similar to the decreased gene expression observed in our tomato pedicel abscission between 16–20 h (Figure 1 and Figure 2, Table 4). A recent report described vesicle trafficking genes that were specifically upregulated in AZ-C of detached mature orange fruits exposed to ethylene for 24 h, using laser microdissection and microarray analysis [47].

Unlike these studies, which focused on one sampling time point during the abscission process, we performed here a gene expression analysis at six time points during the process of tomato pedicel abscission, from induction until the completion of abscission. These time points corresponded to the abscission stages B to D (i.e., abscission induction, early and late stages of abscission execution, and synthesis of the defense layer). In addition, the alternations in gene expression described in the above reports could not be regarded as AZ-specific, as only AZ-rich tissues, but not NAZ tissues, were sampled for sequencing [14,41]. In the experiments described in the present study, samples of both the FAZ and NAZ were analyzed, which enabled to define the genes that are directly related to the abscission process. Thus, our results demonstrate a detailed AZ-specific expression kinetics of cargo- and trafficking-associated genes that are involved in the abscission process in a model system that we had well characterized in previous studies at the whole gene level [9,12,48,49,50].

Although all those reports in this field strongly suggest that vesicle trafficking is involved in abscission, the demonstration of the involvement of the small GTPase NEVERSHED in Arabidopsis floral organ abscission [43,44,45] is the only report in the abscission literature that supports the idea that the loss of function of a specific protein involved in membrane trafficking blocks abscission. Recently, we observed that dis-function of the *Rab1a* (*At5g47200*) using SALK insertion (SALK_117532c) in Arabidopsis resulted in a delayed abscission phenotype (Meir et al., unpublished). The abscission in this SALK mutant appeared at position 14 compared to position 7 in the Col. WT.

Current approaches for separation of the extracellular vesicles (EVs) based on their size have been used to isolate EVs from the apoplast of Arabidopsis leaves [21]. However, this separation technique, based only on the vesicles size, does not allow a clear identification of the type of the isolated vesicles. Additionally, the immuno-purification technique [57] can also be used for specific isolation of the EVs. Using these methods and analyzing the EVs cargo, can shade light on the kinetics of the building up of the defense layer against biotic and abiotic stresses during the formation of the boundary layer, cell separation, and post-separating stages of abscission. 

The endomembrane system, intracellular trafficking network, and exocytosis are very complex and dynamic systems. Therefore, it might be a challenge to track the individual vesicles and their specific cargos in real time. The most direct method of dissecting the role of the vesicle network in their native state, and to identify their constituent cargos, is by vesicle isolation [57,58,59,60]. All these approaches can be used for isolation of components of vesicle secretion in abscission systems, and thereby could enable the study of several vesicle populations in plants in general, and in the AZ in particular. Recently, selected vesicles were isolated using SYNTAXIN OF PLANTS61 (SYP61), as a bait protein on the vesicle surface, for characterizing the polysaccharides cargo directed to the cell wall [61]. Hence, the results of the present research provide a set of target SNAREs for a subsequent detailed study of abscission processes, and the outcoming knowledge might lead to development of new tools for controlling the organ abscission process in various agricultural systems. 

## 3. Materials and Methods

**General statement:** The results presented here on the involvement of vesicles trafficking in the abscission process are unpublished additional results based on the same tomato abscission experiments reported by Kim et al. (2015) [12]. The preparation of the tomato AZ-specific microarray chip and the methods for the transcriptomic analysis were described in detail by Sundaresan et al. (2016) [50].

### 3.1. Plant Material and Abscission Induction Treatments 

The experiments were performed with tomato (*Solanum lycopersicum*, cv. “New Yorker”) plants, grown in 10 L containers in a greenhouse located in The Volcani Center, Israel, under a controlled temperature of 25 °C and natural daylight. Inflorescences bearing at least 2–4 freshly open flowers were harvested from 4-month-old plants, brought to a controlled observation room maintained at 20 °C and 60% to 70% relative humidity with continuous light intensity of 14 μmol m^−2^ s^−1^, and prepared for the pedicel abscission assay as previously described [12]. Pedicel abscission was induced by removing the flowers, which were cut at the base of the receptacle, and evaluated by careful touching the distal side of the FAZ. Tissue samples for RNA extraction, containing 2-mm segments, were taken from the FAZ and the proximal pedicel (NAZ), located 5 mm adjacent to the FAZ. The 2 mm FAZ segments were composed of approximately 1 mm sections located on either side of the AZ joint or fracture. Samples (30 segments for each tissue) were collected at several time points, including 0 h (immediately before flower removal), and at 4, 8, 12, 16, and 20 h after flower removal. Tissue samples were collected from two different inflorescences for two experimental replicates.

### 3.2. RNA Extraction and Microarray Assays

Samples (50 mg) were snap frozen and then homogenized, and RNA isolation and further processing were performed as previously described [50]. The labeled cRNA samples were hybridized onto an AZ-specific microarray chip, AMADID:43310 [50], designed by Genotypic Technology (Pvt. Ltd., Bangalore, India). Each chip included 111,718 probe sets for more than 40,000 transcripts. The probe sets were designed using RNA-seq results for pooled RNA samples from non-induced (control) and induced tomato NAZ and FAZ tissue collected at 2, 4, 8, 12, 16, and 20 h after flower removal as described above. Data extraction from images was performed by using the Agilent Feature Extraction software V-11.5.

Microarray data analysis was performed as previously described [48], using Agilent GeneSpring GX Version 12 software. Duplicate samples were analyzed for each time point, and the duplicated probes within the array were averaged for each given transcript. The expression levels denote the gProcessed signal value from the microarray data (Background subtracted signal intensity values).

### 3.3. Gene Expression Validation by Quantitative Real-Time PCR (qRT-PCR)

Primers for qRT-PCR were designed using Gene Runner version 3.05 (http://www.generunner.net). The primers were validated and amplicon sizes were confirmed using 2% agarose gel as detailed in Appendix A. The primers were designed to match the microarray probes. The RNA samples used for qRT-PCR were the same used for the microarray analysis. Preparation of cDNA and qRT-PCR assay were carried out as previously described [49]. The relative expression levels of the genes were determined after normalizing with *ACTIN* as the reference gene, using the comparative C_T_ method for calculating the value of 2^−∆∆CT^.

### 3.4. Sequence Deposition

The raw sequence data and array information were submitted to the Gene Expression Omnibus (GEO) of the National Center for Biotechnology Information (NCBI) with GEO IDs GSE45355 and array ID AMADID:043310. 

## 4. Conclusions 

We report here, for the first time, the kinetics of the expression of vesicle trafficking-related genes, small GTPases, SNAREs, and SNARE regulators, during abscission, parallel to the expression of genes associated with cell wall disassembling and synthesis of constituents of the boundary and defense layers, that should be excreted to the cell wall and the apoplast for abscission execution. Our results clearly show how these processes are programmed and orchestrated at the level of gene expression. It is a future challenge to relate the expression of the genes associated with cell wall disassembling and synthesis of the boundary and defense layers to the actual vesicles and their cargo. This can be done by isolating different types of vesicles, which contain different abscission-related proteins, at the various time points during the abscission process, and performing their proteomic and metabolomic analyses.

## Figures and Tables

**Figure 1 life-10-00273-f001:**
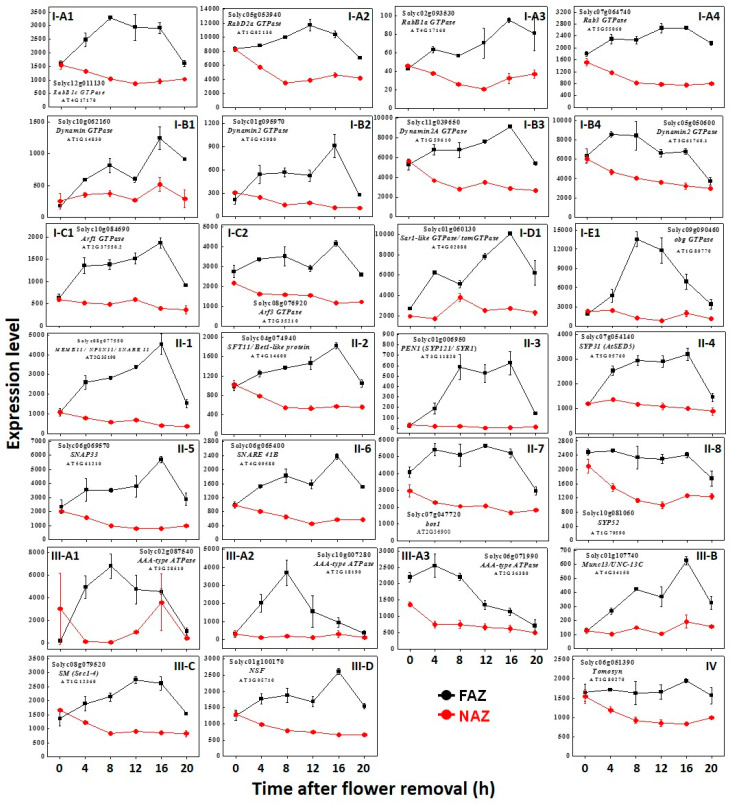
The kinetics of changes in array-measured expression levels of vesicle trafficking-related genes that were upregulated at **4 h** after flower removal in the FAZ (black line) compared to the NAZ (red line) during 20 h after abscission induction. The determination of genes that were upregulated in the FAZ at **4 h** after flower removal was based on comparison to their expression level at zero time (before flower removal). These genes included: (**a**) GTPases (**I**)—*Rab GTPase* (**I-A1**–**A4**), *Dynamin GTPase* (**I-B1**–**B4**), *Arf1 GTPase* (**I-C1**–**C2**), *Sar1-like GTPase* (**I-D1**), and *Obg GTPase* (**I-E1**); (**b**) SNAREs (**II**)—*t-SNAREs* (**II-1**–**II-6,II-8**), and *v-SNAREs* (**II-7**); (**c**) Matchmaker SNARE regulators (**III**)—*AAA-type ATPase* (**III-A1**–**A3**), *Munc13/UNC-13C* (**III-B**), SM—***S****ec1-4/**M**unc* (**III-C**), *NSF* (**III-D**); (**d**) a Matchbreaker SNARE regulator—*Tomosyn* (**IV**). The results are means of three biological replicates ± SD. Transcript identities are indicated by their Solyc and *Arabidopsis thaliana* (AT) gene numbers.

**Figure 2 life-10-00273-f002:**
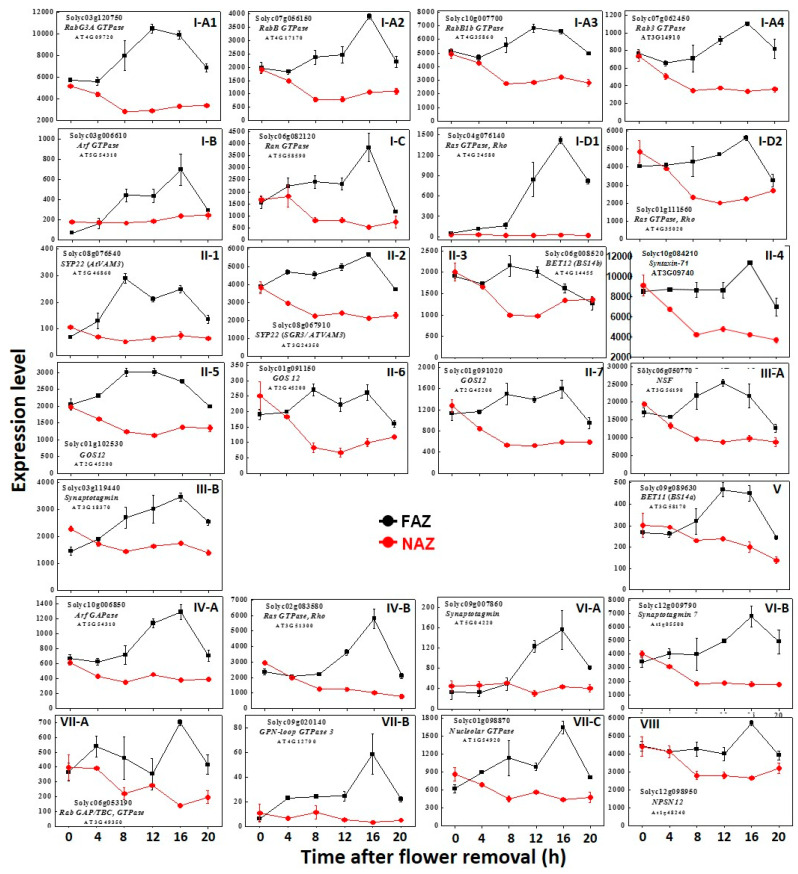
The kinetics of changes in array-measured expression levels of vesicle trafficking-related genes that were upregulated at **8 h** (**I**,**II**,**III**), **12 h** (**IV**,**V**,**VI**) or **16 h** (**VII**,**VIII**) after flower removal in the FAZ (black line) compared to the NAZ (red line) during 20 h after abscission induction. The determination of genes that were upregulated in the FAZ at **8, 12, and 16 h** after flower removal was based on comparison to their expression level at zero time (before flower removal). These genes included: (**a**) GTPases (**I**,**IV**)—*Rab GTPase* (**I-A1-4**), *Arf1 GTPase* (**I-B**,**IV-A**), *Ran GTPase* (**I-C**), and *Ras GTPase* (**I-D**,**IV-B**); (**b**) SNAREs (**II**,**V**,**VIII**)—*t-SNAREs* (**II-1**–**II-4**,**V**,**VII**), and *v-SNAREs* (**II-5**–**II-7**); (**c**) matchmaker SNARE regulators (**III**,**VI**)—*NSF* (**III-A**), and *Synaptotagmin* (**III-B**,**VI-A**,**VI-B**). The results are means of three biological replicates ± SD. Transcript identities are indicated by their Solyc and *Arabidopsis thaliana* (AT) gene numbers.

**Table 1 life-10-00273-t001:** List of changes in the expression of vesicle trafficking-related genes of the three main categories (GTPases, SNAREs, and SNARE regulators) that were significantly upregulated in the FAZ compared to the adjacent NAZ at 4 h after flower removal. The genes were classified according to their functions in the vesicle trafficking process during pedicel abscission at this time point. The vertical arrows ↑ indicate that gene expression in the FAZ at 4 h was upregulated compared to the expression at zero time, and the horizontal arrows → indicate that the gene expression was similar at the two sampling time points. The gene list is based on Figure 1.

Function	GTPases	SNAREs	SNARE Regulators
**Vesicle Budding**	*Arf1 GTPase* Solyc10g084690 ↑ *Arf3 GTPase* Solyc08g076920 ↑ *Dynamin2 GTPase* Solyc01g095970 ↑ *Dynamin2 GTPase* Solyc05g050600 ↑ *Dynamin GTPase* Solyc10g062160 ↑ *Dynamin2A GTPase* Solyc11g039650 ↑		**Matchmakers** *NSF* Solyc01g100170 ↑ *AAA-type ATPase* Solyc06g071990 ↑ *AAA-type ATPase* Solyc02g087540 ↑ *AAA-type ATPase* Solyc10g007280 ↑ *Munc13/UNC-13C* Solyc01g107740 ↑ Sec1-4/Munc *(SM)* Solyc08g079520 ↑
**ER→ cis Golgi**	*RabB1c GTPase* Solyc12g011130 ↑ *RabB1a GTPase* Solyc02g093530 ↑ *RabD2a GTPase* Solyc05g053940 → *Sar1 GTPase* Solyc01g060130 ↑	*bos1—*v-SNARE, Qb-SNARE Solyc07g047720 ↑	
**Golgi Apparatus**		*Bet1* - MEMB11* t-SNARE, Qc- SNARE Solyc04g074940 ↑ *SNARE 11—*SFT11* t-SNARE, Qb-SNARE Solyc08g077550 ↑ *SNARE 41b trans membrane* Solyc06g065400 ↑ *SYP31—*Syntaxin 32 t-SNARE- Qa SNARE Solyc07g054140 ↑	
**Trans Golgi Network (TGN)**			
**TGN→ PM**		*SNAP33*—t-SNARE, Qb+c-SNARE Solyc06g069570 ↑ *PEN1—(SYP121/SYR1)* Syntaxin t-SNARE—Qa SNARE Solyc01g006950 ↑	**Matchbreakers** *Tomosyn* Solyc06g051390 →
**Golgi→Vacuole**		*SYP52—*t- SNARE Syntaxin-51, Qc-SNARE *Solyc10g081060* →	
**Unknown Function**	*obg GTPase* Solyc09g090460 ↑		
**Exocytosis General**	*Rab3 GTPase* Solyc07g064740 ↑		

**Table 2 life-10-00273-t002:** List of changes in the expression of vesicle trafficking-related genes of the three main categories (GTPases, SNAREs**,** and SNARE regulators) that were significantly upregulated in the FAZ compared to the adjacent NAZ **at 8 h** after flower removal. The genes were classified according to their functions in the vesicle trafficking process during pedicel abscission at this time point. The vertical arrows ↑ indicate that gene expression in the FAZ was upregulated at 8 h compared to the expression at 4 h, and the horizontal arrows → indicate that gene expression was similar at the two sampling time points. The gene list is based on Figure 1 and Figure 2. The genes presented in **bold** are those that were specifically upregulated in the FAZ **at 8 h** after flower removal relative to their initial expression levels at zero time before flower removal, as demonstrated in Figure 2I–III.

Function	GTPases	SNAREs	SNARE Regulators
**Vesicle Budding**	*Arf1 GTPase* Solyc10g084690 → *Arf3 GTPase* Solyc08g076920 → *Dynamin2 GTPase* Solyc01g095970 → *Dynamin2 GTPase* Solyc05g050600 → *Dynamin GTPase* Solyc10g062160 ↑ *Dynamin2A GTPase* Solyc11g039650 → ***Arf GTPase*** **Solyc03g006610** ↑		**Matchmakers** *NSF* Solyc01g100170 → *AAA-type ATPase* Solyc06g071990 → *AAA-type ATPase* Solyc02g087540 ↑ *AAA-type ATPase* Solyc10g007280 ↑ *Munc13/UNC-13C* Solyc01g107740 ↑ Sec1-4/Munc *(SM)* Solyc08g079520 → ***NSF*** ***Solyc06g050770*** ↑ ***Synaptotagmin*** *S**olyc03g119440*** ↑
**ER→cis Golgi**	*RabB1c GTPase* Solyc12g011130 ↑ *RabB1a GTPase* Solyc02g093530 → *RabD2a GTPase* Solyc05g053940 ↑ *Sar1 GTPase* Solyc01g060130 → ***RabB GTPase*** ****Solyc07g056150**** ↑ *****RabB1b GTPase***** *****Solyc10g007700***** ↑	*Bos1—*v-SNARE, Qb-SNARE Solyc07g047720 → ***Bet12—*****t-SNARE, Qc-SNARE** **Solyc06g008520 →** ***SYP71*****—t- SNARE Syntaxin**- **Qc-SNARE*****Solyc10g084210*** →	
**Golgi Apparatus, Trans Golgi Network (TGN)**		*Bet1—*MEMB11* t-SNARE, Qc- SNARE Solyc04g074940 → *SNARE 11—*SFT11* t-SNARE, Qb-SNARE Solyc08g077550 → *SNARE41b trans membrane* Solyc06g065400 ↑ *SYP31—*Syntaxin 32 t-SNARE- Qa SNARE Solyc07g054140 ↑ ***GOS12-v-SNARE-C-Qb-SNAR*** **Solyc01g091020** ↑ ***Solyc01g091150*** ↑ **Solyc01g102530** ↑ ***SYP 22*** **-t-SNARE Syntaxin,** **Qa-SNARE Solyc08g076540** ↑ ***SYP22*****-t-SNARE Syntaxin 32-** **Qa-SNARE** ***Solyc08g067910***→	
**TGN→PM**		*SNAP33*-t-SNARE, Qb+c-SNARE Solyc06g069570 → *PEN1*-*(SYP121/SYR1)* Syntaxin t-SNARE-Qa SNARE Solyc01g006950 ↑	**Matchbreakers** *Tomosyn* Solyc06g051390 →
**Golgi→Vacuole**	***RabG3A GTPase*** **Solyc03g120750** ↑	*SYP52*-t- SNARE Syntaxin-51, Qc-SNARE *Solyc10g081060* →	
**Cytoskeleton and Vesicular Trafficking**	***Ras GTPase, Rho*** **Solyc04g076140** ↑ ***Ras GTPase, Rho*** **Solyc01g111560**↑		
**Nuclear Transport**	***Ran GTPase*** **Solyc06g082120** →		
**Unknown Function**	*obg GTPase* Solyc09g090460 ↑		
**Exocytosis General**	*Rab3 GTPase* Solyc07g064740 → ***Rab3 GTPase*** **Solyc07g062450** →		

**Table 3 life-10-00273-t003:** List of changes in the expression of vesicle trafficking-related genes of the three main categories (GTPases, SNAREs, and SNARE regulators) that were significantly upregulated in the FAZ compared to the adjacent NAZ at **12 h** after flower removal. The genes were classified according to their functions in the vesicle trafficking process during pedicel abscission at this time point. The vertical arrows ↑, ↓ indicate that gene expression in the FAZ at 12 h was upregulated or downregulated, respectively, compared to the expression at 8 h, and the horizontal arrow → indicates that gene expression was similar at the two sampling time points. The gene list is based on Figure 1 and Figure 2. The genes presented in **bold** are those that were specifically upregulated in the FAZ **at 12 h** after flower removal relative to their initial expression levels at zero time before flower removal, as demonstrated in Figure 2IV–VI.

Function	GTPases	SNAREs	SNARE Regulators
**Vesicle Budding**	*Arf1 GTPase* Solyc10g084690 → *Arf3 GTPase* Solyc08g076920 → *Dynamin2 GTPase* Solyc01g095970 → *Dynamin2 GTPase* Solyc05g050600 → *Dynamin GTPase* Solyc10g062160 → *Dynamin2A GTPase* Solyc11g039650 → *Arf GTPase* Solyc03g006610 → ***Arf-GAP*** ***Solyc10g006850*** ↑		**Matchmakers** *NSF* Solyc01g100170 → *AAA-type ATPase* Solyc06g071990 ↓ *AAA-type ATPase* Solyc02g087540 ↓ *AAA-type ATPase* Solyc10g007280 ↓ *Munc13/UNC-13C* Solyc01g107740 → Sec1-4/Munc *(SM)* Solyc08g079520 ↑ *NSF* *Solyc06g050770* ↑ *Synaptotagmin* *Solyc03g119440* → ***Synaptotagmin*** **Solyc09g007860** ↑ ***Synaptotagmin** 7* **Solyc09g007860** →
**ER→cis Golgi**	*RabB1c GTPase* Solyc12g011130 → *RabB1a GTPase* Solyc02g093530 → *RabD2a GTPase* Solyc05g053940 ↑ *Sar1 GTPase* Solyc01g060130 ↑ *RabB GTPase* Solyc07g056150 → *RabB1b GTPase* *Solyc10g007700* ↑	*Bos1—*v-SNARE, Qb-SNARE Solyc07g047720 → *Bet12—*t-SNARE, Qc-SNARE Solyc06g008520 → *SYP71*-t- SNARE Syntaxin- Qc-SNARE *Solyc10g084210* → ***BET11—*****t-SNARE, Qc-SNARE** **Solyc09g089630** ↑	
**Golgi Apparatus, Trans Golgi Network (TGN)**		*Bet1—*MEMB11 t-SNARE, Qc- SNARE Solyc04g074940 → *SNARE 11—*SFT11 t-SNARE, Qb-SNARE Solyc08g077550 → *SNARE41b trans membrane* Solyc06g065400 ↑ *SYP31—*Syntaxin 32 t-SNARE- Qa SNARE Solyc07g054140 → *GOS12 -v-SNARE-C-Qb-SNAR* Solyc01g091020 → *Solyc01g091150* → Solyc01g102530 → *SYP32*—t-SNARE Syntaxin 32- Qa-SNARE *Solyc08g067910* → *SYP 22*-t-SNARE Syntaxin, Qa-SNARE Solyc08g076540 ↓	
**TGN→PM**		*SNAP33*—t-SNARE, Qb+c-SNARE Solyc06g069570 → *PEN1—(SYP121/SYR1)* Syntaxin t-SNARE—Qa SNARE Solyc01g006950 →	**Matchbreakers** *Tomosyn* Solyc06g051390 →
**Golgi→Vacuole**	*RabG3A GTPase* Solyc03g120750 ↑	*SYP52—*t- SNARE Syntaxin-51, Qc-SNARE *Solyc10g081060* →	
**Cytoskeleton and Vesicular Trafficking**	*Ras GTPase, Rho* Solyc04g076140 ↑ *Ras GTPase, Rho* Solyc01g111560 → ***Ras GTPase, Rho*** **Solyc02g083580** ↑		
**Nuclear Transport**	*Ran GTPase* Solyc06g082120 →		
**Unknown Function**	*obg GTPase* Solyc09g090460 →		
**Exocytosis General**	*Rab3 GTPase* Solyc07g064740 ↑ *Rab3 GTPase* Solyc07g062450 ↑		

**Table 4 life-10-00273-t004:** List of changes in the expression of vesicle trafficking-related genes of the three main categories (GTPases, SNAREs and SNARE regulators) that were significantly upregulated in the FAZ compared to the adjacent NAZ **at 16 h** after flower removal. The genes were classified according to their functions in the vesicle trafficking process during pedicel abscission at this time point. The vertical arrows ↑, ↓ indicate that gene expression in the FAZ was upregulated or downregulated, respectively, compared to the previous sampling time point (12 h), and the horizontal arrows → indicate that gene expression was similar in the two sampling time points. The gene list is based on Figure 1 and Figure 2. The genes presented in **bold** are those that were specifically upregulated in the FAZ **at 16 h** after flower removal relative to their initial expression levels at zero time before flower removal, as demonstrated in Figure 2VII-A–C,VIII.

Function	GTPases	SNAREs	SNARE Regulators
**Vesicle Budding**	*Arf1 GTPase* Solyc10g084690 ↑ *Arf3 GTPase* Solyc08g076920 ↑ *Dynamin2 GTPase* Solyc01g095970 ↑ *Dynamin2 GTPase* Solyc05g050600 → *Dynamin GTPase* Solyc10g062160 ↑ *Dynamin2A GTPase* Solyc11g039650 ↑ *Arf GTPase* Solyc03g006610 ↑ *Arf-GAP* *Solyc10g006850* ↑		**Matchmakers** *NSF* Solyc01g100170 ↑ *AAA-type ATPase* Solyc06g071990 ↓ *AAA-type ATPase* Solyc10g007280 ↓ *Munc13/UNC-13C* Solyc01g107740 ↑ Sec1-4/Munc *(SM)* Solyc08g079520 ↑ *NSF* *Solyc06g050770* ↓ *Synaptotagmin* *Solyc03g119440* → *Synaptotagmin* Solyc09g007860 → *Synaptotagmin 7* Solyc09g007860 ↑
**ER→cis Golgi**	*RabB1c GTPase* Solyc12g011130 → *RabB1a GTPase* Solyc02g093530 ↑ *RabD2a GTPase* Solyc05g053940 ↓ *Sar1 GTPase* Solyc01g060130 ↑ *RabB GTPase* Solyc07g056150 ↑ *RabB1b GTPase* *Solyc10g007700* →	*Bos1—*v-SNARE, Qb-SNARE Solyc07g047720 → *SYP71*-t- SNARE Syntaxin- Qc-SNARE *Solyc10g084210* ↑ *BET11—*t-SNARE, Qc-SNARE Solyc09g089630 →	
**Golgi Apparatus, Trans Golgi Network (TGN)**		*Bet1—*MEMB11* t-SNARE, Qc- SNARE Solyc04g074940 ↑ *SNARE 11—*SFT11* t-SNARE, Qb-SNARE Solyc08g077550 ↑ *SNARE41b trans membrane* Solyc06g065400 ↑ *SYP31—*Syntaxin 32 t-SNARE- Qa SNARE Solyc07g054140 → *GOS12-v-SNARE-C-Qb-SNAR* Solyc01g091020 → *Solyc01g091150* → Solyc01g102530 ↓ *SYP32—*t-SNARE Syntaxin 32- Qa-SNARE *Solyc08g067910* → *SYP22*-t-SNARE Syntaxin, Qa-SNARE Solyc08g076540 →	
**TGN→PM**		*SNAP33*—t-SNARE, Qb+c-SNARE Solyc06g069570 ↑ *PEN1—(SYP121/SYR1)* Syntaxin t-SNARE-Qa SNARE Solyc01g006950 → ***NPSN12—*****t-SNARE Qb-SNARE** **Solyc12g098950** ↑	**Matchbreakers** ***Tomosyn*** **Solyc06g051390** ↑
**Golgi→Vacuole**	*RabG3A GTPase* Solyc03g120750 →	*SYP52—*t-SNARE Syntaxin-51, Qc-SNARE *Solyc10g081060* →	
**Cytoskeleton and Vesicular Trafficking**	*Ras GTPase, Rho* Solyc04g076140 ↑ *Ras GTPase, Rho* Solyc01g111560 ↑ *Ras GTPase, Rho* Solyc02g083580 ↑		
**Nuclear Transport**	*Ran GTPase* Solyc06g082120 ↑		
**Unknown Function**	*obg GTPase* Solyc09g090460 ↓		
**Exocytosis General**	*Rab3 GTPase* Solyc07g064740 → *Rab3 GTPase* Solyc07g062450 ↑	+3 new GTPases	

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
