# Peer review of "Expression Kinetics of Regulatory Genes Involved in the Vesicle Trafficking Processes Operating in Tomato Flower Abscission Zone Cells during Pedicel Abscission"

_life, 2020, doi:10.3390/life10110273_

Round 1

Reviewer 1 Report

The secretion of various cell wall enzymes is required for organ abscission, but compared to its importance, the regulatory mechanism has not been identified. This paper by Sundaresan et al. reports the expression kinetics of vesicle trafficking components including SNAREs, SNARE regulators and small GTPases providing clues for how the processes of protein secretion are regulated. Abscission was induced by flower removal and time-dependent gene expression was analyzed by microarray comparing pedicel abscission zone (FAZ) and the proximal pedicel (NAZ). These data will provide the basis for understanding how the secretion of cell wall enzymes is regulated in the future.

Comments:

- It is important to validate microarray data. I strongly recommend to perform qRT-PCR to confirm DEG information obtained with microarray.

- Rather than graphing individual gene expression patterns, it is better to group those showing similar patterns into a graph, then graph the kinetics of each group and display the expression of genes belonging to each group through a heat map. In addition, if the gene expression of the vesicle trafficking component was analyzed along with the gene expression of the cargo, it would provide more valuable information. Analysis of cell wall enzymes that show expression kinetics similar to vesicle trafficking components can provide clues as to how the trafficking machinery and cargo are regulated.

- What makes this paper different from other papers is that gene expression was analyzed by dividing it into six time points. In addition to the vesicle trafficking genes, if there is anything that has not been revealed in previous studies at the whole gene level, it would be good to explain this to maximize its benefits.

Reviewer 2 Report

Review of "Expression Kinetics of Regulatory Genes Involved in the Vesicle Trafficking Processes Operating in Tomato Flower Abscission Zone Cells During Pedicel Abscission" by Sundaresan et.

This paper is a continuation of the work by the research group that published several crucial works on abscission. The main object is the flower pedicel of tomato, but the findings apply to a broader understanding of the molecular processes driving abscission. The research is based on a transcriptome analysis of a previously used abscission transcriptome time series dataset. Research of exo-/endocytosis in abscission is rare, so it is very important to have a transcriptomic analysis of the genes involved in membrane transport processes as a foundation for further research. Degradation of middle lamella and loosening of the cell wall is a key step in execution of abscission. Effectors have to be delivered to the separation layer, therefore it is of great interest to research membrane transport processes during abscission.

The topic is well introduced with the relevant works properly cited. There are some research papers that describe vesicular membrane structures forming in the cells of the abscission zone and may be related to vesicle based membrane tranfer. If appropriate, you may include them in the introduction or discussion. Gilliland et al. (1976, https://doi.org/10.1002/j.1537-2197.1976.tb13173.x) observed delivery of acid phosphatase rich vesicles that fuse with plasma membrane in the separation zone in hibiscus petal AZ. Accumulation of vesicular structures was observed in proximal part of tomato leaf abscission zone by Bar-Dror et al. (2011, https://doi.org/10.1105/tpc.111.092494) and in tomato flower AZ by Chersicola et al. (2017, https://doi.org/10.3389/fpls.2017.00464). In yellow lupin flower AZ, vesicles in cytoplasm were observed (Wilmowicz et al. 2018, https://doi.org/10.1007/s10725-018-0375-7).

Graphs in figures are well presented and readable. Missing in the legend is what is Expression level? Describe briefly in the legend or in the methods how is expression quantified.

The information in Tables 1, 2, 3 and 4 is informative and needed, but the tables are not readable in the present form. Can the "SNARE regulators" column be moved to the bottom of the table (as a row)?

Minor edits:
line 374: the GEO accession GSE45356 contains leaf AZ data, so I believe it is not needed to cite it here.
line 378, 382: dissembling -> disassembling

I would support publication of this paper with minor modifications.

Round 2

Reviewer 1 Report

The authors have answered my previous concerns adequately.